# Evaluation of 8-Channel Radiative Antenna Arrays for Human Head Imaging at 10.5 Tesla

**DOI:** 10.3390/s21186000

**Published:** 2021-09-08

**Authors:** Myung Kyun Woo, Lance DelaBarre, Matt Thomas Waks, Young Woo Park, Russell Luke Lagore, Steve Jungst, Yigitcan Eryaman, Se-Hong Oh, Kamil Ugurbil, Gregor Adriany

**Affiliations:** 1Center for Magnetic Resonance Research, University of Minnesota, Minneapolis, MN 55455, USA; wooxx@umn.edu (M.K.W.); dela0087@umn.edu (L.D.); waks0005@umn.edu (M.T.W.); park1556@umn.edu (Y.W.P.); rllagore@umn.edu (R.L.L.); jungs001@umn.edu (S.J.); yigitcan@umn.edu (Y.E.); ugurb001@umn.edu (K.U.); 2Department of Electrical and Computer Engineering, Seoul National University, Seoul 08826, Korea; 3Department of Biomedical Engineering, Hankuk University of Foreign Studies, Yongin 17035, Korea; ohs@hufs.ac.kr

**Keywords:** center-fed antenna, dipole antenna, end-fed antenna, monopole antenna, sleeve antenna, ultra-high field imaging

## Abstract

For human head magnetic resonance imaging at 10.5 tesla (T), we built an 8-channel transceiver dipole antenna array and evaluated the influence of coaxial feed cables. The influence of coaxial feed cables was evaluated in simulation and compared against a physically constructed array in terms of transmit magnetic field (B_1_^+^) and specific absorption rate (SAR) efficiency. A substantial drop (23.1% in simulation and 20.7% in experiment) in B_1_^+^ efficiency was observed with a tight coaxial feed cable setup. For the investigation of the feed location, the center-fed dipole antenna array was compared to two 8-channel end-fed arrays: monopole and sleeve antenna arrays. The simulation results with a phantom indicate that these arrays achieved ~24% higher SAR efficiency compared to the dipole antenna array. For a human head model, we observed 30.8% lower SAR efficiency with the 8-channel monopole antenna array compared to the phantom. Importantly, our simulation with the human model indicates that the sleeve antenna arrays can achieve 23.8% and 21% higher SAR efficiency compared to the dipole and monopole antenna arrays, respectively. Finally, we obtained high-resolution human cadaver images at 10.5 T with the 8-channel sleeve antenna array.

## 1. Introduction

Fundamentally, the physics of magnetic resonance imaging (MRI) dictates that the achievable signal-to-noise ratio (SNR) increases with magnetic field strength [1,2,3,4]. Hence, ultra-high field (UHF) MRIs at 7 tesla (T) and above can yield higher resolution images or faster acquisition times over conventional high-field MRIs of 1.5 T or 3 T. Currently, a 10.5 T/447 MHz MRI system at University of Minnesota is the highest fully operational whole human body MRI scanner and is at the forefront of the MRI hardware technology [5,6,7,8]. Since the wavelength at 447 MHz (in the presence of human tissue) is within the dimension of the imaging subject, unique challenges, particularly related to overall field homogeneity, arise at UHF, and the design of radiofrequency (RF) coil arrays for MRI applications benefits from serious consideration of antenna concepts [9,10]. This is a significant change from the strict near-field regime-dominated RF coil arrays operating at clinical MRI frequencies below 3 T/128 MHz. At UHF frequencies, radiative-type antennas, particularly dipole antennas, have been suggested as excellent building blocks for transmit arrays and have, indeed, shown promising performance [11,12]. Compared to other coil types, such as loop [13,14] or microstrip antennas [15,16], dipole antennas have the additional advantage of symmetric B_1_^+^ field (defined as the transmit magnetic field generated by an RF coil) patterns, as well as a favorable direction of energy propagation (Poynting vector), which results in greater penetration depth. Consequently, both dipole antennas and the combination of loops with dipole antennas have been successfully used for UHF human imaging applications with high penetration, efficiency, and SNR [17,18,19]. However, for realistic head array housings, the coaxial feed cables for dipole antennas have to be routed in close proximity to one leg of the dipole and they interact with the antenna elements.

In a recent publication [8], we had started to evaluate alternative antenna arrays and compared 16-channel dipole and sleeve antenna arrays. Since intra-element coupling due to the tight spacing of the 16 channels can negatively affect the overall antenna array performance, for the study presented here, we compared more sparsely spaced 8-channel arrays. For an 8-channel dipole antenna head array, we first compared an ‘ideal’ coil (Figure 1a) without coaxial cables with a realistic array, which included coaxial cables (Figure 1c), in simulations, and evaluated the effect of coaxial feed cables on the dipole antenna performance. To investigate the influence of this effect experimentally, we built an 8-channel dipole antenna head array and compared two setups: one with a more ideal—but, in practice, unrealistic—coaxial feed cable path (Figure 1e) and the other with a realistic close-proximity coaxial feed cable path (Figure 1g). This allowed us to compare the performance of the 8-channel dipole antenna array with a defined phantom load, both in electromagnetic (EM) simulations and 10.5 T MR experiments with the phantom. To evaluate alternative antenna layouts with possibly more advantageous coaxial feed points for head imaging, we built two end-fed antenna arrays: an 8-channel monopole antenna array [20,21] and an 8-channel sleeve antenna array [8]. Then, we compared the B_1_^+^ efficiency (defined as B_1_^+^ field normalized by the square root of net input power) and specific absorption rate (SAR) efficiency (defined as B_1_^+^/√(peak 10 g SAR)) of the 8-channel dipole, monopole, and sleeve antenna arrays with the phantom. Finally, we compared the performance of these arrays with a human model. 

## 2. Materials and Methods

### 2.1. Simulation and Numerical Analysis

Investigating the influence of coaxial cables in detail required that the 8-channel inductor-shortened dipole antenna array was modeled without (Figure 1a) and with (Figure 1c) coaxial cables. EM simulations were performed for both arrays, and the B_1_^+^ efficiency (Figure 1b,d,f,h)), 10 g SAR (Figure 2a,b), and SAR efficiency (Figure 2c,d) maps were compared between the two models. Then, for the comparison with 8-channel end-fed antenna arrays, 8-channel monopole (Figure 3a) and sleeve antenna (Figure 3b) arrays were modeled with coaxial cables. B_1_^+^ efficiency, 10 g SAR, and SAR efficiency maps of the monopole and sleeve antenna arrays were calculated and compared with the phantom.

A cylindrical phantom with ε_r_ = 49 and σ = 0.6 S/m was used to approximate human brain tissue properties for the performance comparison of the arrays. The diameter (=18 cm) and the length (=30.5 cm) of the phantom was chosen to emulate the comparable size of a human head and neck.

The performance of the modeled arrays (dipole, monopole, and sleeve antenna) was compared using a human model (Duke) [22]. To closely resemble the practical coil setup, in which the coaxial feed cables are inherently part of the coil, the antenna models included the coaxial cables and the cable traps. The simulated B_1_^+^ efficiency and SAR efficiency were calculated and compared among the three antenna arrays. B_1_^+^ efficiency, 10 g SAR, and SAR efficiency of each array were summarized in Table 1.

Simulated B- and E-fields were obtained using XFdtd (REMCOM, State College, PA, USA). By driving each of the eight elements, with relative transmit phase increasing by 45 degrees with each channel, the net B_1_ field generated by the array was in a circularly polarized mode. The portion of the total RF magnetic field that can induce spin excitation, the B_1_^+^ fields, were obtained with the equation below:(1)$$B_{1}^{+}=\left|\frac{B_{x}+iB_{y}}{2}\right|,$$
where B_x_ and B_y_ are the complex amplitudes of the x- and y-oriented RF magnetic fields, respectively [23]. Based on the EM simulation, B_1_^+^ efficiency and SAR efficiency maps were calculated with MATLAB (The Mathworks, Inc., Natick, MA, USA). For quantitative comparison, the highest B_1_^+^ efficiency, 10 g SAR, and SAR efficiency areas were obtained from the region of interest (ROI) in the axial plane for all arrays. The related values obtained from 2 mm isotropic voxel ROI are indicated below each of the axial images. Red dotted lines of each coronal plane indicate the location of the axial plane and display of B_1_^+^ efficiency. Red arrows indicate ROIs where these B_1_^+^ efficiency values are measured.

### 2.2. Construction of 8-Channel Dipole, Monopole, and Sleeve Antenna Arrays

The coil formers for the dipole and sleeve antenna arrays were modeled (Fusion 360, Autodesk, Mill Valley, CA, USA) and then 3D printed (F410, Fusion3 Design, Greensboro, NC, USA). The dipole and sleeve antenna arrays have the same elliptically shaped physical dimensions, with a short and long axis of 20 cm × 22 cm and a length of 20 cm with eight equally spaced antenna elements distanced at 8.8 ± 1.4 cm between individual elements. The individual dipoles and sleeve antennas of the 8-channel dipole and sleeve antenna arrays were tuned to 447 MHz with wound inductors. In Figure 3c, the monopole antenna array was built with a 24 cm × 24 cm dimension and a length of 20 cm on a circular-shaped acrylic former [20,21]. Eight quarter-wave (~13 cm at 10.5 T) monopole antennas were connected to a 35 cm × 36 cm dimension copper ground plate, which was segmented and reconnected with 330 pF capacitors to reduce eddy currents. As shown in our previous research [6,8,17,19], all arrays were operated without an intra-element decoupling circuit.

A lattice balun with two ceramic capacitors (10 pF, 100B series, American Technical Ceramics, Huntington Station, NY, USA) and two inductors (12 nH, A04T_L, Coilcraft Inc., Cary, IL, USA) was used for all matching networks of the 8-channel dipole antenna array. Lattice balun circuits at the feed point and one floating cable trap on the feed cable reduced the sheath current for each element of the 8-channel dipole antenna array. For the comparison in the experiment, the coaxial cables of the dipole antenna array were positioned as a further and tight fit setup. The coaxial feed cable path of the 8-channel dipole antenna array was adjusted to maximize distance from the resonant structure to reduce the interaction between the coaxial cables and elements of the array.

The sleeve antenna array was basically designed with the combination of monopole antennas and floating cable traps [24,25]. To support a better comparison with the inductor-shortened dipole antenna array, the length of the sleeve antenna array elements was also shortened with inductors. For the sleeve antenna array [8], the length of each monopole conductor of the antenna was set to 15 cm with inductor shortening, and combined with a 5 cm floating cable trap for an overall length of 20 cm. For the dipole and sleeve antenna arrays, similar floating cable traps were equipped on the coaxial cables [8,26,27,28].

Appendix A summarizes bench measurements obtained from each of the arrays and indicates the range of reflection coefficient (S_11_) and coupling coefficient (S_21_) values when the arrays are loaded with a cylindrical phantom. All S_11_ and S_21_ were measured in bench measurements using a 16-channel network analyzer (ZNBT8, Rohde & Schwarz, Munich, Germany). The S-parameters of each channel of the 8-channel dipole, monopole, and sleeve antenna arrays were measured in dB-scaled values.

### 2.3. Experimental Setup

A Siemens MAGNETOM console equipped with 16 independent transmit and 32 receive channels (Siemens, Erlangen, Germany) was used in conjunction with a Siemens SC72 body gradient coil inside the 88 cm diameter whole-body 10.5 T magnet (Agilent, Santa Clara, CA, USA). All data presented here were acquired using parallel transmit (pTx) with equal RF transmit power per channel. An in-house designed 8-channel transmit/receive interface (Virtumed, Minneapolis, MN, USA) with 25 dB receiver gain was used to connect the arrays to the MR scanner. The phantom with exactly the same physical dimensions and electrical properties of the simulation was used to compare the performance of arrays experimentally. The phantom was filled with sucrose, saline, and distilled water solution and measured with the DAKS-12 probe (SPEAG, Zurich, Switzerland) to confirm electrical properties [29]. The phantom was carefully positioned within each of the coil formers to achieve agreement between simulation and experiment and mimic the location for human brain imaging.

For the basic experimental antenna element coupling evaluation of all antenna arrays, noise covariance matrices were obtained in a 10.5 T MR experiment, as illustrated in Appendix A. Noise covariance matrices of the dipole antenna with further (Appendix A) and close (Appendix A) coaxial setup, the monopole antenna (Appendix A), and sleeve antenna (Appendix A) arrays were acquired to experimentally evaluate the crosstalk between the elements [30]. Transmit B_1_^+^-field efficiency maps of individual elements were obtained using an actual flip-angle imaging (AFI) sequence (TR_1_/TR_2_ = 25/115 ms, TE = 3.39 ms, nominal flip angle = 60°, GRAPPA (R = 2), and resolution = 2 mm × 4 mm × 6 mm) with the cylindrical phantom. The flip angle with short TR_1_ and TR_2_ was calculated by:(2)$$\alpha =\arccos\;\frac{\left(rn-1\right)}{\left(n-r\right)},$$
where α = flip angle and n = TR_2_/TR_1_, and r ≈ $\frac{1+n\;cos\;\alpha }{n+\cos\alpha }$ [31]. The flip angle was converted to B_1_^+^ with:α = 2πγ B_1_^+^ τ,(3)
where τ is the width in seconds of the RF pulse [32].

To demonstrate overall imaging capabilities and brain coverage, we obtained high-resolution human cadaver images using B_1_-phase shimming with the 8-channel sleeve antenna array at 10.5 T. High-resolution human cadaver images were obtained for the evaluation of the 8-channel sleeve antenna array utilizing B_1_-phase shimming (defined as phase optimization using a pTx system). Turbo spin-echo (TSE) images (TR = 5000 ms, TE = 72 ms, TA = 15:18 min, BW = 488 Hz/pixel, NT = 4, FOV = 200 mm × 159 mm, and resolution = 0.39 mm × 0.39 mm × 1.0 mm) and T_1_-weighted gradient-recalled echo (GRE) images (TR = 150 ms, TE = 3.4 ms, TA = 4:58 min, BW = 421 Hz/pixel, NT = 4, FOV = 200 mm × 200 mm, and resolution = 0.3 mm × 0.3 mm × 1.8 mm) were acquired with the human cadaver.

## 3. Results

### 3.1. Evaluation of Interaction among Coaxial Cables and Antenna Elements with the Inductor-Shortened Dipole Antenna Array

As shown in Figure 1 and Figure 2, we evaluated the interaction among coaxial cables and antenna elements with an 8-channel inductor-shortened dipole antenna array in simulation and experiment. A good agreement between simulation (Figure 1b,f) and experiment (Figure 1d,h) of ~10% was achieved for the highest B_1_^+^ efficiency values. In simulation, the 8-channel dipole antenna array without coaxial cables (Figure 1b) produced 23.1% higher B_1_^+^ efficiency compared to the dipole antenna array with coaxial cables (Figure 1d). The experimental B_1_^+^ efficiency of the dipole antenna array with further (=7 cm gap) coaxial setup arrays was shown to be 20.7% higher compared to the tight (=2 cm gap) coaxial cable setup, as shown in Figure 1f,h. Degradation of the B_1_^+^ efficiency is the disadvantage from the interference between coaxial cables and antenna elements. However, SAR efficiency values of the dipole antenna array were similar between without (Figure 2c) and with (Figure 2d) coaxial cables, because SAR values (Figure 2a,b) were also decreased as B_1_^+^ efficiency values were decreased.

### 3.2. Comparison of the 8-Channel Monopole and Sleeve Antenna Arrays

As shown in Figure 4, we compared the 8-channel monopole and sleeve antenna arrays to evaluate the end-fed antenna arrays. Figure 4 shows the simulated (Figure 4a,b)) and experimental (Figure 4c,d) B_1_^+^ efficiency comparison between the monopole and sleeve antenna arrays in the central axial and the coronal planes. Within the ROI in Figure 4, the 8-channel monopole antenna array displayed 24.4% and 30.4% higher B_1_^+^ efficiency than the 8-channel sleeve antenna array in the simulation and the experiment, respectively.

The expected peak 10 g SAR values within the phantom of the 8-channel monopole (Figure 4e) and sleeve (Figure 4f) antenna arrays are 0.47 W/kg and 0.28 W/kg, respectively. Compared to the 8-channel monopole antenna array, the results indicate 40.4% higher peak 10 g SAR values of the 8-channel sleeve antenna array. Due to the lower peak 10 g SAR values of the 8-channel sleeve antenna array, both the monopole (Figure 4g) and the sleeve (Figure 4h) antenna arrays showed similar SAR efficiency values.

### 3.3. Comparison of the Dipole, Monopole, and Sleeve Antenna Arrays with Human Model and Human Cadaver Images with the Sleeve Antenna Array

When the simulated human model data were compared within the ROI in Figure 5a–c), the 8-channel sleeve antenna array displayed similar B_1_^+^ efficiency to the 8-channel dipole antenna array, and 22% lower B_1_^+^ efficiency than the 8-channel monopole antenna array. However, due to the lower peak 10 g SAR values, the 8-channel sleeve antenna array (Figure 5f) showed 23.8% higher and 21% higher SAR efficiency compared to the 8-channel dipole antenna (Figure 5d) and monopole antenna (Figure 5e) arrays, respectively.

The sleeve antenna array showed relatively small coverage in the phantom (Figure 4b,d). However, with the human model, the sleeve antenna (Figure 5c) did not show smaller B_1_^+^ coverage compared to the dipole (Figure 5a) nor the monopole (Figure 5b) antenna arrays.

Figure 6 shows coronal high-resolution TSE and GRE cadaver images obtained at 10.5 T with the 8-channel sleeve antenna array, which demonstrate overall field distribution and coverage. Uniform whole-brain images were achieved with the 8-channel sleeve antenna array utilizing B_1_ phase shimming technique, while maintaining equal RF amplitude per channel. Optimized pTx excitation techniques can further improve homogeneity and local B_1_^+^ efficiency.

## 4. Discussion

We focused on the following three points in this work. First, we evaluated how much degradation was present in an 8-channel dipole antenna array due to the interference among coaxial cables and antennas. Minimizing feed cable interferences is an important point, particularly for UHF head applications, due to the possibility for variance in the distance of the antenna element to the head tissue, which can result in possible stronger interaction of the antenna element with the coaxial feed cable than the imaging object. The second important point is a careful comparison among three different types of 8-channel head antennas. The third point is to demonstrate a close match between the simulation results and the physical experiment. It is important since the accurate simulation of RF coils with human models is ultimately used to estimate the RF coil safety at UHF [6].

The inductor-shortened dipole antenna array—albeit with a practically unrealistic coaxial feed cable routing (Figure 1e,f)—showed higher B_1_^+^ efficiency values compared to more practical tight cable routing (Figure 1g,h). This distant feed cable setup utilized for phantom imaging is not practical in most clinical research setups, and a substantial B_1_^+^ efficiency drop was observed with a more realistic coaxial feed cable setup. In this respect, the monopole and sleeve antenna arrays have a significant advantage in that the coaxial feed cables are routed away from the patient while keeping the performance of antennas optimal.

In our initial 16-channel research [8], the asymmetric sleeve antenna array concept showed lower peak 10 g SAR and higher SAR efficiency compared to a dipole antenna array. Besides the benefits of the balanced coaxial feed point, this is the result of the principal advantage of the sleeve antenna concept with the important degree of freedom to alter the geometry of the sleeve parts, as well as monopole antenna parts. This allows for a practical compromise between the high E- and B-fields close to the ground plane of classical monopole arrays (Figure 4) and the more advantageous field patterns of dipoles for central slices. The ability of a balanced array feed point positioned away from the tissue/load results in a unique advantage of the sleeve antenna concepts for head applications. As an outcome, in the head array research presented here, we observed reduced peak 10 g SAR for the sleeve antenna array compared to the dipole and monopole antenna arrays. Generally, the feed point of an antenna has high current flow and produces strong E-fields in the imaging sample. In recent 10.5 T dipole antenna work by Sadeghi-Tarakameh [6], this issue was elegantly addressed by increasing the physical distance of the dipole antenna feed point to the sample, resulting in what the author termed a ‘bumped’ dipole antenna. Similarly, the E-fields spread out around the feed points of the sleeve antenna array, since the feed points of the individual antennas were pulled away from the subject for the experimental setup. Consequently, the asymmetric sleeve antenna array has an increased region of 10 g SAR towards the feed point (Figure 4f); however, this spreading out of 10 g SAR distribution leads to lower peak 10 g SAR and higher SAR efficiency values compared to the inductor-shortened dipole antenna array.

The monopole antenna array showed the highest B_1_^+^ efficiency with the phantom and the human model compared to the dipole and sleeve antenna arrays close to the ground plane. However, it showed relatively higher peak 10 g SAR with the human model compared to the phantom. Based on comparison among Figure 2, Figure 4 and Figure 5, SAR efficiency of each array was consistently lowered with the human model compared to the phantom. The SAR efficiency of the phantom/human model was 0.91/0.8 (13.8% drop) with the dipole antenna array, 1.2/0.83 (44.6% drop) with the monopole antenna array, and 1.18/1.05 (12.4% drop) with the sleeve antenna array. Due to the nonuniform distance from the antenna to the human subject, SAR efficiency can differ between a human model and a phantom. The field-generating ‘monopole’ antenna parts of both the monopole and the sleeve antennas are expected to perform similarly; however, the classical monopole antenna is using a pronounced ‘RF ground plane’ as an electrical current mirror antenna element. This results in clear differences in E- and B-fields between the monopole and sleeve antenna arrays.

The low peak 10 g SAR leads to the highest SAR efficiency of the sleeve antenna array (Figure 5f) over the ROI compared to the dipole (Figure 5d) and the monopole (Figure 5e) antenna arrays. The highest SAR efficiency in the human model is a particularly important advantage of the asymmetric sleeve antenna array concept compared to both dipole and monopole antenna arrays. The sleeve antenna array has an asymmetric structure that is configured with monopole antennas (i.e., 15 cm) and sleeves (i.e., 5 cm). E-fields of a sleeve antenna change depending on the size of the sleeve. For the sleeve antenna concept, we have the freedom to modify both the geometry of the sleeve parts and the monopole antenna parts.

As shown in Appendix A, without any decoupling circuitry, each 8-channel array produced acceptable decoupling values and noise covariance values to proceed with the imaging experiment. However, the coupling and noise covariance of the 8-channel sleeve antenna array (Appendix A) showed the lowest overall values compared to the 8-channel dipole (Appendix A) and monopole (Appendix A) antenna arrays. It is an additional advantage of the sleeve antenna concept.

## 5. Conclusions

We evaluated three types of 8-channel radiative antenna arrays for UHF MRI. Among them, the sleeve antenna array showed clear advantages in terms of lower peak 10 g SAR and higher SAR efficiency compared to the dipole and monopole antenna arrays. Finally, the incorporation of a sleeve into the feed structure has significant positive practical implications, and the variation in sleeve length adds an important degree of design freedom.

The sleeve antenna concepts can also support tighter element spacing, and thus, potentially, could allow for increased density of antenna elements in array configurations. The deliberate lower element counts of arrays presented in this work support better comparison of head-sized dipole and monopole antenna arrays, and they represent an important validation step. For future in vivo human brain imaging research with antenna arrays, the sleeve antenna concept can be further expanded towards a higher number of elements through, for example, dual row layouts and variations in the length of individual antennas. Eventually, we plan to explore combinations of sleeve transmitters with dedicated receiver inserts as an important step towards higher channel array configuration in excess of 32 receive elements. Furthermore, incorporation of high-permittivity materials in support of shorter elements will be considered in addition to further improved geometry arrangements for SAR optimization.

The future experiments will also utilize enhanced B_1_ field control techniques using pTx with individual control of the phase, amplitude, and RF pulse shape of each coil element and the possibility for significantly improved transmit homogeneity.

## Figures and Tables

**Figure 1 sensors-21-06000-f001:**
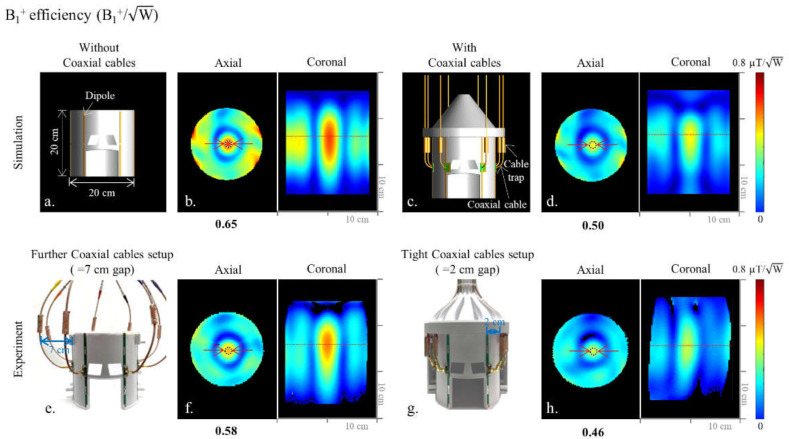
CAD models of the 8-channel inductor-shortened dipole antenna array without (**a**) and with (**c**) coaxial cable setups. Corresponding simulated B_1_^+^ efficiency maps of the dipole antenna array are shown in (**b**,**d**), respectively. Photographs of the 8-channel inductor-shortened dipole antenna array with further (**e**) and tight (**g**) coaxial cable setups. Corresponding experimental B_1_^+^ efficiency maps of the dipole antenna array are shown in (**f**,**h**), respectively.

**Figure 2 sensors-21-06000-f002:**
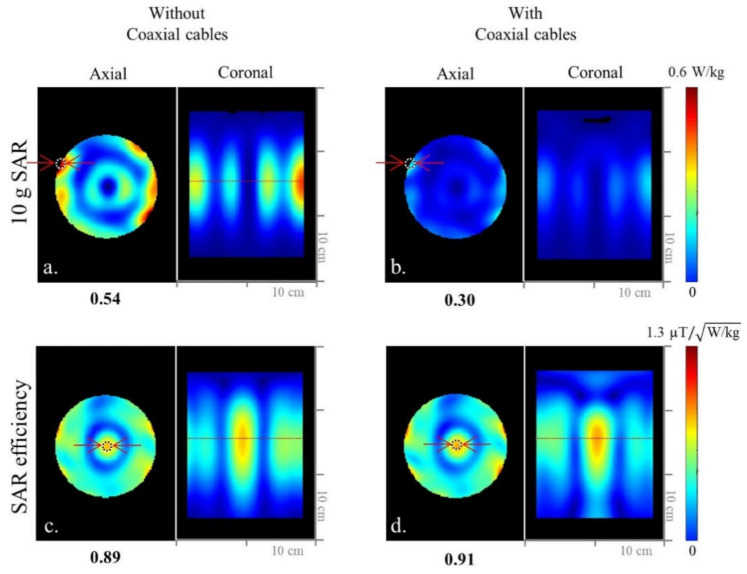
Simulated 10 g SAR (**a**,**b**) and SAR efficiency (**c**,**d**) maps of the 8-channel inductor-shortened dipole antenna array without and with coaxial cable setups in the axial and coronal planes.

**Figure 3 sensors-21-06000-f003:**
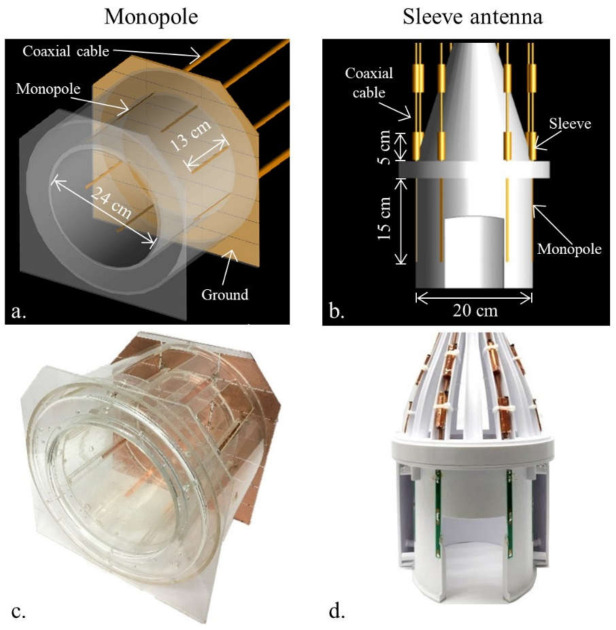
CAD models (**a**,**b**) and photographs (**c**,**d**) of the 8-channel monopole and sleeve antenna arrays.

**Figure 4 sensors-21-06000-f004:**
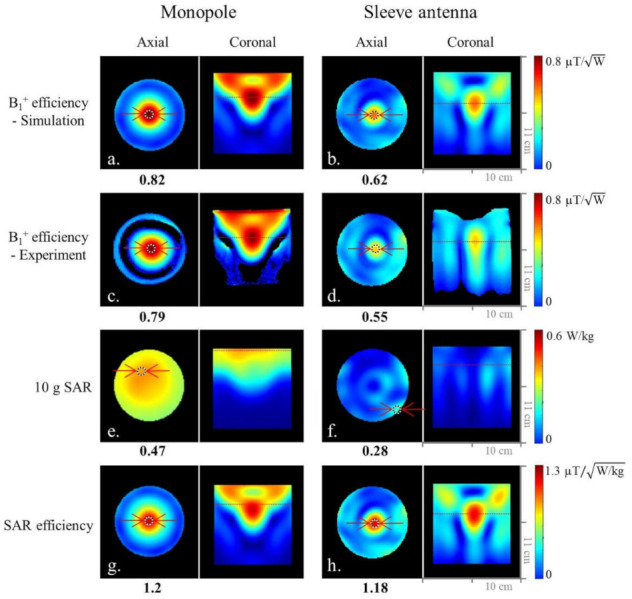
Simulated (**a**,**b**) and experimental (**c**,**d**) B_1_^+^ efficiency maps of the 8-channel monopole and sleeve antenna arrays in the axial and coronal planes. Simulated 10 g SAR (**e**,**f**) and SAR efficiency (**g**,**h**) maps of the 8-channel monopole and sleeve antenna arrays in the axial and coronal planes.

**Figure 5 sensors-21-06000-f005:**
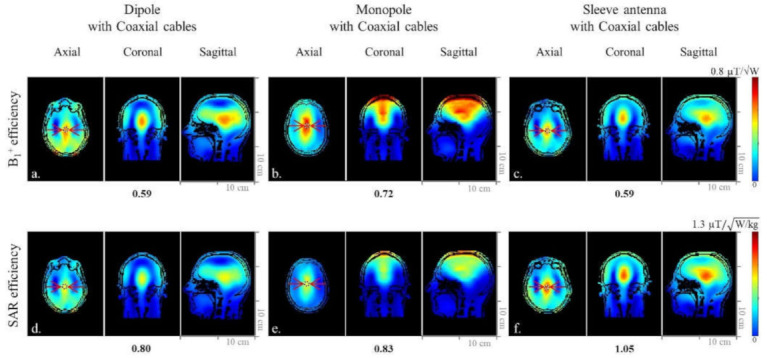
Simulated B_1_^+^ efficiency (**a**–**c**) and SAR efficiency (**d**–**f**) maps of the 8-channel dipole, monopole, and sleeve antenna arrays with coaxial cables in the axial, coronal, and sagittal planes.

**Figure 6 sensors-21-06000-f006:**
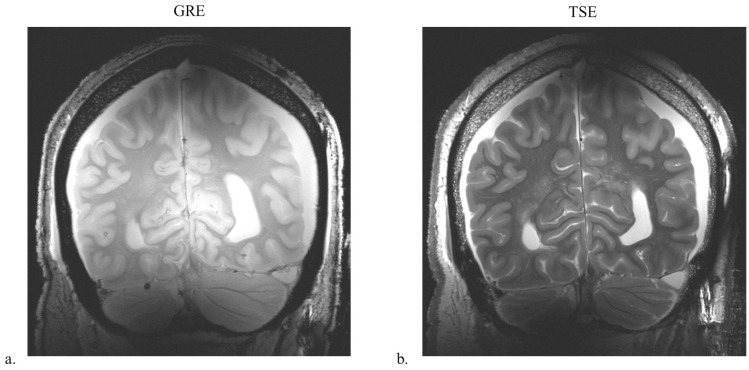
B_1_-phase-shimmed GRE (**a**) and TSE (**b**) images of the 8-channel sleeve antenna array with human cadaver in the coronal plane.

**Table 1 sensors-21-06000-t001:** Quantitative comparison of B_1_^+^ efficiency, 10 g SAR, and SAR efficiency among the 8-channel dipole, monopole, and sleeve antenna arrays with human model in the ROI marked for the highest value.

	Dipole	Monopole	Sleeve Antenna
B_1_^+^ efficiency (μT/√W)	0.59	0.72	0.59
Peak 10 g SAR (W/kg)	0.55	0.74	0.31
SAR efficiency (µT$/\sqrt{W/\mathrm{kg}}$)	0.80	0.83	1.05

## Data Availability

The data presented in this study are available on request from the corresponding author. The data are not publicly available due to the restriction of local law and government policy.

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
