# Peer review of "Evaluation of 8-Channel Radiative Antenna Arrays for Human Head Imaging at 10.5 Tesla"

_sensors, 2021, doi:10.3390/s21186000_

Round 1

Reviewer 1 Report

Please, find my comments in the attached document.

Reviewer 2 Report

This paper proposes the validity of the MRI application of the proposed sleeve antenna structure based on the simulation and measurement results. 

However, the structure or terminology of the manuscript is difficult for readers to understand and there are some logically inconsistent parts, so I think it needs to be corrected. The parts to be improved are as follows. 

  1. Why is it important to do this work at 10.5 T? I want to know why.
  2. From the point of view of medical applications, I would like you to explain what the size of each indicator (B1+ efficiency, SAR efficiency) should be and why.
  3. What is the term ROI in line 81? 
  4. The structure of the manuscript is difficult to understand. It is necessary to change the configuration logically at a glance. For example, it would be good to remove Chapter 2 and continue the explanation according to the figures in sequence. In Chapter 2, foretelling the figure that follows is confusing the reader. 
  5. In Figure 3, mark the antenna part and the cable part separately. 
  6. Supplementary Table 1 is not shown on line 125. Please include. The other supplementary figures mentioned are also not visible. 
  7. In the sentence below on lines 173-175 "In simulation, the 8-channel dipole antenna array without coaxial cables (Figure 1(a)) produced 23.1 % lower B1+ efficiency compared to dipole antenna array with coaxial cables (Figure 1(b))". Shouldn't it be the other way around? As you can see from the figure, the case without cables is 23.1% more efficient. 
  8. in the 179-181th sentences "SAR efficiency values of the dipole antenna array were similar between without (Figure 2(c)) and with (Figure 2(d)) coaxial cables because SAR values (Figure 2(a) and 2(b)) were also decreased as B1+ efficiency values were decreased." So what's the conclusion? Is the influence of the cable not a big issue? Please explain the meaning of the experimental results.
  9. It would be better to explain clearly what is the difference between the experimental set-ups in Figures 4 and 5, and explain why the SAR efficiency of the sleeve antenna array is improved in Figure 5 unlike Figure 4. 

Round 2

Reviewer 1 Report

All comments were addressed. For these reasons I recommend the paper for publication.

Author Response

Thank you for your review.

Reviewer 2 Report

I believe that the authors generally faithfully answered the points raised and reflected them in the revised paper. However, as a final conclusion, the 8-channel sleeve type array antenna showed the best performance, but it seems that the analysis on why this is the case is insufficient. It was explained that the performance of the dipole type array antenna deteriorates due to the coupling between the actual cables, but for the sleeve type, the final simulation result (Fig. In Figure 4, there is no big difference from monopole, but it is better in Figure 5, and I think that analysis of the cause of the reason, that is, why the 10 g SAR is significantly smaller than other types of antennas, should be added. 

Author Response

Thank you for your indication.

In our initial publication [Ref #10], we demonstrated why a sleeve antenna array can show lower E-fields compared to a dipole antenna array based on the theory and analysis with the simulation. We didn’t emphasize too much the asymmetric structure of a sleeve antenna in this manuscript. As we described in the method section, the sleeve antenna array has an asymmetric structure which is configured with 15 cm long monopole antennas and 5 cm long sleeves. E-fields of a sleeve antenna change depending on the size of a sleeve. It is an advantage of the sleeve antenna concept that we have the freedom to modify both the geometry of the sleeve parts and the monopole antenna parts. Based on King et al, the asymmetric sleeve antenna does not show a sine functional current distribution - as the dipole antenna does - as long as the monopole side of antenna is longer than (λ/4). 

The sleeve antenna can resemble a monopole antenna. Theoretically, the monopole antenna part of the sleeve antenna performs similar to that of a monopole antenna using an electrical mirror-image as an antenna element. From Antenna System Guide [Ref #17] we referred, that it is relatively easy to intuitively verify how a sleeve antenna resembles a monopole antenna. As shown in Figure 4., there is a similar pattern of overall B1+ efficiency between the monopole and sleeve antenna arrays. However, a major difference was shown at the superior part of the phantom. In the superior part of the phantom the classical monopole antenna array - which has a physically big ground plane - shows significant high B1+ and E-fields compared to the sleeve antenna array. This is also shown with the human model. We added a sentence to addressing your previous comment, E-field and SAR efficiency can differ between a human model and a phantom due to the non-uniform distance from the antenna to the human subject.

We now added further statements in the discussion section of this manuscript.

- R. King, “Asymmetrically driven antennas and the sleeve dipole,” Proc. IRE, vol. 38, no. 10, pp. 1154–1164, Oct. 1950.